# Phagocytic Activities of Reactive Microglia and Astrocytes Associated with Prion Diseases Are Dysregulated in Opposite Directions

**DOI:** 10.3390/cells10071728

**Published:** 2021-07-08

**Authors:** Anshuman Sinha, Rajesh Kushwaha, Kara Molesworth, Olga Mychko, Natallia Makarava, Ilia V. Baskakov

**Affiliations:** 1Center for Biomedical Engineering and Technology, University of Maryland School of Medicine, Baltimore, MD 21201, USA; sanshuman@som.umaryland.edu (A.S.); rajkushwaha@som.umaryland.edu (R.K.); kmolesworth@som.umaryland.edu (K.M.); omychko@som.umaryland.edu (O.M.); nmakarava@som.umaryland.edu (N.M.); 2Department of Anatomy and Neurobiology, University of Maryland School of Medicine, Baltimore, MD 21201, USA

**Keywords:** prions, prion diseases, neuroinflammation, reactive microglia, reactive astrocytes, phagocytosis

## Abstract

Phagocytosis is one of the most important physiological functions of the glia directed at maintaining a healthy, homeostatic environment in the brain. Under a homeostatic environment, the phagocytic activities of astrocytes and microglia are tightly coordinated in time and space. In neurodegenerative diseases, both microglia and astrocytes contribute to neuroinflammation and disease pathogenesis, however, whether their phagocytic activities are up- or downregulated in reactive states is not known. To address this question, this current study isolated microglia and astrocytes from C57BL/6J mice infected with prions and tested their phagocytic activities in live-cell imaging assays that used synaptosomes and myelin debris as substrates. The phagocytic uptake by the reactive microglia was found to be significantly upregulated, whereas that of the reactive astrocytes was strongly downregulated. The up- and downregulation of phagocytosis by the two cell types were observed irrespective of whether disease-associated synaptosomes, normal synaptosomes, or myelin debris were used in the assays, indicating that dysregulations are dictated by cell reactive states, not substrates. Analysis of gene expression confirmed dysregulation of phagocytic functions in both cell types. Immunostaining of animal brains infected with prions revealed that at the terminal stage of disease, neuronal cell bodies were subject to engulfment by reactive microglia. This study suggests that imbalance in the phagocytic activities of the reactive microglia and astrocytes, which are dysregulated in opposite directions, is likely to lead to excessive microglia-mediated neuronal death on the one hand, and the inability of astrocytes to clear cell debris on the other hand, contributing to the neurotoxic effects of glia as a whole.

## 1. Introduction

Chronic neuroinflammation is considered as one of the major pathological manifestations of neurodegenerative diseases including Alzheimer’s disease, Parkinson’s disease, Amyotrophic Lateral Sclerosis, and prion diseases [1]. In prion diseases, both astrocytes and microglia undergo profound transcriptional, morphological, and functional transformation resulting in reactive phenotypes [2,3,4,5]. Reactive phenotypes of astrocytes and microglia involve the global dysregulation of their homeostatic functions that manifest in the disturbances of multiple physiological pathways [2,3].

Both microglia and astrocytes contribute to neuroinflammation. However, their reactive states appear to play opposite roles in disease pathogenesis. Mounting evidence suggests that reactive astrocytes associated with prion diseases are neurotoxic and perhaps even drive the disease pathogenesis [3,6,7,8]. Reactive astrocytes isolated from prion-infected animals were synaptotoxic to primary neurons [6]. The selective, astrocyte-specific targeting of the unfolded protein response, which is exuberated in reactive astrocytes, was found to slow down the rate of disease progression [7]. The degree of astrocyte activation along with the disturbance in their physiological pathways inversely correlated with the incubation time to prion disease suggesting that the reactive states contribute to the rate of disease progression [3]. In contrast to astrocytes, reactive microglia appear to be neuroprotective and have a net positive impact [9,10,11,12]. Partial ablation of microglia by PLX5622 exacerbated the reactive astrocyte phenotype and accelerated disease progression [5]. Elimination of three microglia-derived factors TNF-α, IL-1α, and C1qa was sufficient to accelerate the progression of prion diseases [13].

Microglia are regarded as the main cells in CNS responsible for phagocytosis, yet astrocytes are capable of phagocytic uptakes too [14,15,16]. In fact, under homeostatic conditions, phagocytic activities of astrocytes and microglia are tightly coordinated in time and separated in space, as each cell type plays a specialized role in the removal of damaged neurons and apoptotic bodies [14]. The questions whether, in prion diseases, phagocytic activities of reactive microglia and astrocytes are up- or down-regulated, and whether the changes are unidirectional, have not been examined.

In prion diseases, and, in particular, in animals infected with the ME7 prion strain, microglia were engaged in the phagocytosis of beads or apoptotic cells but did not effectively remove PrP^Sc^ [17]. On the contrary, microglia and astrocyte cell lines cultured in vitro were effective in phagocytic clearance of PrP^Sc^ [18,19,20]. Moreover, colocalization of PrP^Sc^ with reactive microglia and/or astrocytes in prion-infected animals suggest that microglia and astrocytes preserve their phagocytic activities in their reactive states [21,22,23,24]. However, it remains unknown whether the phagocytic activity of glia cells is up- or down-regulated in their reactive states in comparison to their homeostatic states.

To address the above question, this current study isolated microglia and astrocytes from C57BL/6J mice infected with prions and tested their phagocytic activities toward synaptosomes and myelin debris in live-cell imaging assays. We found that the reactive microglia upregulate, whereas the reactive astrocytes downregulate phagocytic uptakes. The up- and downregulation of phagocytosis by reactive microglia and astrocytes, respectively, were observed regardless of whether disease-associated synaptosomes, normal synaptosomes, or myelin debris were used in phagocytic assays. These results indicate that the dysregulations are dictated by the reactive phenotype of glia, but not by the substrates. Dysregulation of the phagocytic activities, which occurred in opposite directions for microglia and astrocytes, is likely to disable clearance of cell debris and contribute to the neurotoxic effects of glia.

## 2. Materials and Methods

### 2.1. Reagents, Kits, Antibodies

Sodium bicarbonate, Poly-D-lysine (PDL), Poly-L-lysine (PLL), protease inhibitor cocktail (PIC), tween 20, bovine serum albumin (BSA), paraformaldehyde (PFA), horse serum, normal goat serum (NGS), triton-X-100, CellLytic MT mammalian cell lysis buffer, dimethyl sulfoxide (DMSO), and ponceau S were purchased from Sigma Chemical Co. (St. Louis, MO, USA). Dulbecco’s modified eagle medium: F12 (DMEM/F12), neurobasal medium, Trypsin-EDTA, Hank’s balanced salt solution (HBSS), phosphate buffer saline (PBS), trypsin inhibitor, antibiotic-antimycotic, Dulbecco’s phosphate-buffered saline (DPBS), glutamax, protein ladder, and fetal bovine serum (FBS) were procured from Invitrogen (Carlsbad, CA, USA). The adult mouse brain dissociation kit, cell debris removal solution, myelin removal solution, LS column, C-tubes, and RBC lysis solution were procured from Miltenyi Biotec (Bergisch Gladbach, Germany). SuperSignal West Femto Maximum Sensitivity Substrate was procured from Thermo Scientific (Rockford, IL, USA) and VECTASHIELD mounting medium with DAPI was purchased from Vector Laboratories (Burlingame, CA, USA). SYBR Green, the iScript cDNA Synthesis Kit, and the Aurum Total RNA Mini Kit were procured from Bio-Rad laboratories (Hercules, CA, USA). 0.22 µm filter, 70 μm nylon mesh filter, polyvinylidenefluoride (PVDF) membrane, and Bicinchoninic Acid (BCA) protein assay kit were procured from Millipore (Temecula, CA, USA). Transferrin, putrescine, progesterone, sodium selenite, N-acetyl cysteine (NAC), and Heparin-binding EGF-like growth factor (HBEGF) were purchased from Sigma-Aldrich (St. Louis, MO, USA).

The following antibodies were used: horseradish peroxidase (HRP), conjugated secondary anti-mouse IgG (cat. A9044), anti-rabbit IgG (cat. A0545), mouse monoclonal antibodies to β-actin (cat. A5441), chicken polyclonal antibody to GFAP (cat. AB5541) were all from Sigma-Aldrich (St. Louis, MO, USA); mouse monoclonal antibody to PSD-95 (cat. 75028020, Antibodies Incorporated, Davis, CA, USA); rabbit monoclonal antibody to GFAP (cat. 12389, Cell Signaling Technology, Danvers, MA, USA); rabbit polyclonal anti-Iba1 antibody (cat. 01919741, Wako, Richmond, VA, USA); rabbit monoclonal antibodies to synaptophysin (cat. ab32594), Olig2 (cat. ab136253), S100β (cat. ab52642) and CD11b (cat. ab133357) were from Abcam (Cambridge, MA, USA); mouse monoclonal antibody to NeuN (cat. mab377, Millipore, Temecula, CA, USA); Alexa Fluor 488 goat anti-chicken IgG, Alexa Fluor 488 goat anti-rabbit IgG, Alexa Fluor 488 goat anti-mouse IgG, Alexa Fluor 546 goat anti-mouse IgG, and Alexa Fluor 546 goat anti-rabbit IgG secondary antibodies were from Invitrogen (Carlsbad, CA, USA).

### 2.2. Animals

Mice were housed in a 12 h day-and-night cycle environment with ad libitum availability of food and water. Six-week-old C57BL/6J males and females were intraperitoneally inoculated with a 200 µL volume of 1% 22L brain homogenate in PBS under anesthetic conditions. The age-matched male control groups were intraperitoneally injected with PBS (200 µL volume) only. Animals were regularly observed and scored for the neurological signs and disease progression. Mice were scored and euthanized when they showed consistent progression of the diseases (developed kyphosis, became lethargic, or could no longer ambulate) at 176–238 days post-inoculation. Upon euthanasia, brains were dissected out, then cortexes were used for primary cultures, and whole brains were used for synaptosome isolation. Similarly, for the isolation of primary astrocytes, SSLOW-infected C57BL/6J mice were euthanized upon showing the consistent progression of the diseases (112–127 days post-inoculation) [21,25,26]. Utilized in the study were 132 animals (60 22L-infected; 60 aged-matched controls; 6 SSLOW-infected; 6 aged-matched controls). For the immunohistochemistry of the brains and Nanostring analysis, C57BL/6J mice were inoculated with a 1% 22L brain homogenate in PBS 22L intracranially, as described before [3].

### 2.3. Isolation and Culturing of Adult Primary Astrocytes

Adult primary cortical astrocyte cultures were prepared using 22L-infected and age-matched control C57BL/6J mice, as well as SSLOW-infected and aged-matched control C57BL/6J mice according to the following protocol [6]. Briefly, the skin was opened at the midline of the head using microdissection scissors, the skull was cut at the midline fissure, and the brain was released from the cavity of the skull. One brain was used per individual culture. A brain was carefully moved to a 60 mm petri dish and was rinsed with cold DPBS to remove the adhering blood. After the removal of the meninges, the brain cortices were dissociated and digested using a papain-based enzyme dissociation solution and incubated for 30 min on the gentleMACS Octo Dissociator device (Miltenyi Biotec) as per the manufacturer’s instructions. Cells were resuspended after digestion in a buffer containing DPBS, 100 U/mL penicillin, and 100 μg/mL streptomycin, after which non-cellular debris was removed by passing the cell suspension through a single-cell strainer of 70 μm nylon. The cell suspension was then centrifuged at 1000 rpm for 10 min. The pellet obtained was incubated for 10 min at 4 °C with myelin removal solution and centrifuged at 1000 rpm for 5 min. The pellet obtained was re-suspended in a complete astrocyte growth media [DMEM/F12 (containing 1mM sodium pyruvate and 365 μg/mL L-glutamine), 100 U/mL penicillin, 100 µg/mL streptomycin and 10% heat-inactivated FBS]. Cells were then seeded at a plate density of 3–4 × 10^4^ per well of chamber slides or 7 × 10^5^ per flask on poly-L-lysine (PLL)-coated chamber slides/cover slips or culture flasks and grown at 37 °C with 5% CO_2_ in a humidified CO_2_ incubator. Total media was changed with fresh media to eliminate the unattached dead cells and debris the next day after plating the cells. The medium was fully replaced after incubating primary astrocyte cultures for 6–7 days. Flasks were covered in plastic, kept in a horizontal position on a shaker platform with the medium covering the cells, and were shaken to remove microglia and oligodendrocytes from the culture at 150 rpm for 1 h at 37 °C. Astrocyte-enriched cultures were washed with PBS immediately after shaking, fresh astrocyte growth culture medium was added and cultured until confluence (2–3 weeks). The medium of culture was replaced every two days. Co-immunostaining of GFAP (astrocyte marker) with NeuN, Iba1, and Olig2 was performed to confirm the purity of the astrocyte cultures.

### 2.4. Isolation and Culturing of Adult Primary Microglia

Adult primary microglia cultures were prepared using clinically ill 22L-infected or age-matched control mice as previously described [6]. Cortical tissue was separated and kept in cold DPBS. One brain was used per individual culture. Meninges were carefully removed from the cortices, then the tissues were transferred to an enzyme mixture prepared using an adult mouse brain dissociation kit (Miltenyi Biotec) and incubated on a rotated gentleMACS Octo Dissociator system for 30 min according to the manufacturer’s instructions. Digested cells were re-suspended in DPBS and filtered through a 70 μm single-cell strainer followed by centrifugation for 10 min at 1000 rpm at 4 °C. The pellets were re-suspended in 30% isotonic Percoll (GE Healthcare, Chicago, IL, USA), and PBS (1X) was gently superimposed on the cell suspension. Percoll gradient solution was then centrifuged for 30 min without breaks at 4000 rpm and 4 °C. The upper portion containing myelin and astrocytes was carefully aspirated. The pellet was dissolved in the RBC lysis buffer for the removal of residual red blood cells (RBC) and incubated at 4 °C for 6–8 min, then the solution was centrifuged for 5 min at 2000 rpm at 4 °C. The pellet obtained was re-suspended in growth media [DMEM/F12 (containing 1 mM sodium pyruvate and 365 µg/mL L-glutamine), 100 U/mL penicillin, 100 µg/mL streptomycin and 10% heat-inactivated FBS]. Cells were then seeded at a plate density of 7 × 10^5^ cells per flask or 3–4 × 10^4^ per well on PDL-coated culture flasks or chamber slides and grown at 37 °C with 5% CO_2_ in a humidified CO_2_ incubator until they become confluent (10–20 days). The medium of culture was replaced every two days. Co-immunostaining of CD11b (microglia marker) with NeuN, GFAP, and Olig2 was performed to confirm the purity of the microglial cultures.

### 2.5. Synaptosome Purification

Synaptosomes were purified from adult 22L-infected and age-matched control mouse brains using the protocol by Loh and co-authors with slight modifications [27]. All steps were carried out at 4 °C. Mouse brains were homogenized individually in 4 mL of the buffer 1 (5 mM HEPES [pH 7.4], 0.5 mM CaCl_2_, 1 mM MgCl_2_, 0.32 M sucrose, and 1 mM DTT supplemented with protease inhibitors) using the Dounce homogenizer with 6–7 up–down strokes. The suspension, labeled as the ‘homogenate’ or H fraction, was cleaned by centrifugation at 4000 rpm for 10 min, and the supernatant was collected. The pellet (P1) was resuspended in 20 mL Buffer 1, then the suspension was centrifuged at 2500 rpm for 10 min and the supernatant was combined with the supernatant collected after the first centrifugation and collectively labeled as S1 fraction. S1 was further fractionated by centrifugation at 13,000 rpm for 10 min, and the supernatant (soluble cytoplasmic fraction or S2) was collected for analysis. The resulting pellet (P2) was resuspended in 5 mL of the buffer 2 (6 mM Tris (pH 8.1), 1 mM EDTA, 0.32 M sucrose, 1 mM DTT, and 1 mM EGTA supplemented with protease inhibitors). A sucrose gradient consisted of the following layers 3 mL of each (from bottom to top) in 15 mL QuickSeal tubes (Beckman Coulter, Brea, CA, USA): 1.2 M, 1.0 M, and 0.85 M sucrose, each in 6 mM Tris (pH 8.1). The fraction P2 was layered over the sucrose gradient and centrifuged at 29,000 rpm for 1.5 h. Fractions at the interface of the 1.0 and 1.2M sucrose layers were collected, diluted with 15 mL of the buffer 2, and centrifuged at 13,000 rpm for 30 min producing the final pellet referred to as Synaptosomes. Synaptosomes were kept frozen at −80 °C. For isolating myelin debris, extra-synaptic fraction (E) with myelin debris formed at the 0.85 M sucrose layer was collected and centrifuged at 13,000 rpm for 30 min. The formed pellets were kept frozen at −80 °C until use.

### 2.6. Transmission Electron Microscopy

Isolated synaptosomes (syn) suspensions were fixed with 2.0% glutaraldehyde in 0.1 M sodium phosphate buffer (pH 7.4; Electron Microscopy Sciences) at room temperature for 2 h and then centrifuged for 10 min at 14,000 rpm to attain a working pellet. At room temperature, pellets were treated with 50 mM glycine in 0.1 M PIPES buffer (pH 7.4) and enrobed in agarose. Agarose pieces containing isolated synaptosomes were then post-fixed with 1.0% osmium tetroxide and 0.25% potassium ferrocyanide in 0.1 M PIPES buffer (pH 7.4) for 60 min at 4 °C. After washing with water, specimens were dehydrated using the following solutions in series: 30% ethanol, 50% ethanol, 70% ethanol containing 1% uranyl acetate, 90% ethanol, and 100% ethanol. Specimens were then incubated with two changes of 100% acetone and infiltrated, in increasing concentrations of Araldite-Epoxy resin (Araldite, Embed 812; Electron Microscopy Sciences, PA, USA), and embedded in pure resin at 60 °C for 24 to 48 h. Ultrathin sections at 70 nm thickness were cut on Leica UC6 ultramicrotome (Leica Microsystems, Inc., Bannockburn, IL, USA), and examined in a Tecnai T12 transmission electron microscope (Thermo Scientific, Hillsboro, OR, USA) operated at 80 kV. Digital images were acquired by using an AMT bottom mount CCD camera and AMT600 software (Advanced Microscopy Techniques, Woburn, MA, USA). The study was performed at the Electron Microscopy Core Imaging Facility (EMCIF) which is part of the Center for Innovative Biomedical Resources (CIBR) and is jointly sponsored by the School of Dentistry, the School of Medicine, and the Marlene and Stewart Greenbaum Cancer Center, University of Maryland, Baltimore.

### 2.7. pHrodo Red-Conjugation

The synaptosomes isolated from the 22L-infected or age-matched control mice were centrifuged at 13,000 rpm for 4 min at 4 °C. The supernatant was discharged; 100 µL of 0.1 M Na_2_CO_3_ and 1 µL of pHrodo Red dye were added to the pellet of synaptosomes (0.3 mg), then the tubes were mildly vortexed two times and covered with aluminum foil to block out the light. The tubes were kept on a twist shaker at 40 rpm for 1–2 h at room temperature. Then, 1 mL of ice-cold DPBS was added, the tubes were centrifuged at 13,000 rpm for 1–2 min at 4 °C, and the supernatant was discharged. The pellet was resuspended in 1 mL of DPBS by pipetting, the suspension was again centrifuged, and these steps were repeated six times to fully remove any unbound pHrodo Red dye. At the end of washing, the supernatant was completely removed, then 100 µL of DPBS with 5% DMSO was gently added to the pellet and the pellet was resuspended by pipetting. Similarly, myelin debris isolated and collected during synaptosome purification was conjugated with pHrodo Red dye. Aliquots of the pHrodo Red-conjugated synaptosomes or myelin were stored at −80 °C.

### 2.8. Preparation of Phagocytosis Assay Medium

Phagocytosis assay medium was prepared by mixing 62.5 mL of Neurobasal medium, 62.5 mL of DMEM/F-12 medium, 1.25 mL of penicillin-streptomycin, 1.25 mL of 100 mM sodium pyruvate, 1.25 mL of 200 mM L-glutamine, 1.25 mL of 100X SATO [800 mg of transferrin, 800 mg of bovine serum albumin, and 128 mg putrescine in 80 mL of Neurobasal medium + 20 µL of progesterone stock (2.5 mg progesterone in 100 µL of 100% ethanol) + 800 µL of sodium selenite stock (4 mg sodium selenite + 10 µL of 1 M NaOH in 10 mL of Neurobasal medium)) and 20 µL of NAC (N-acetyl cysteine; 5 mg/mL), and then filtered through a 0.22 µm filter and aliquoted into 50 mL conical tubes. Stock solution (10 µg/mL) of heparin-binding EGF-like growth factor (HB-EGF) was prepared and stored as aliquots at −80 °C. HB-EGF was freshly mixed with the phagocytosis media (final concentration: 5 ng/mL) just before starting the phagocytosis assay.

### 2.9. Live Imaging Phagocytic Assay with pHrodo Red-Conjugated Synaptosomes or Myelin Debris

An equal density of microglia or astrocyte cells isolated from animals were plated in PDL-coated 96-well or 24-well plates and cultured until 70–80% confluency. The cell media was removed from each well, then wells were washed twice with DPBS, then 300 µL of phagocytosis assay medium containing 2 µL of pHrodo Red-conjugated synaptosomes or pHrodo Red-conjugated myelin debris was added into each well. The plates were kept in the incubator with 5% CO_2_ at 37 °C for 40 min, then the phagocytosis assay media was removed, and the wells were washed thrice with DPBS to remove any unbound pHrodo Red-conjugated synaptosomes/myelin debris. Then, 500 µL of fresh phagocytosis assay medium without synaptosomes/myelin debris was added into each well, then imaging was conducted using Incucyte S3 Live-Cell Analysis System (Essen Bioscience Inc., Ann Arbor, MI, USA) for 24 h. The phagocytic index was calculated by dividing the mean red area (pHrodo Red) by phase area confluency (total glial cell) using Incucyte software (Incucyte 2020B) as previously described [28].

### 2.10. Confocal Microscopy Imaging of Phagocytic Uptake

Astrocytes isolated from animals were plated into the chamber slides (3 × 10^4^ per well) (Thermo Fisher Scientific, Waltham, MA, USA) and cultured until 70–80% confluency, then treated with a phagocytosis assay medium and pHrodo Red-conjugated synaptosomes as described above, but instead of Incucyte Live instrument, these chamber slides were incubated in a cell culture incubator with 5% CO_2_ at 37 °C, for 24 h. The media was removed, the chamber slides were washed thrice with DPBS, then 300 µL of 4% PFA (paraformaldehyde) was added into each well and incubated at room temperature for 20 min. PFA was removed and the wells were washed twice with 1X PBS, then 300 µL of blocking buffer containing 1% NGS and 3% BSA in PBS was added into each well and incubated at room temperature for 2 h. Primary anti-PSD-95 antibody (1:500, 300 µL/well) was incubated overnight at 4 °C, then the wells were washed five times with PBST and incubated with the secondary antibody Alexa Fluor 488 (1:500, 300 µL/well) for 2 h on the shaker at 15 rpm covered with aluminum foil. After washing five times with PBST, cells were mounted with cover glass using Vectashield mounting medium containing DAPI. Images were taken using a Leica confocal microscope SP8 (Leica Microsystems Inc., Buffalo Grove, IL, USA).

### 2.11. Lysotracker Assay

Primary microglia or astrocytes were seeded in a two-well chambered coverglass (ThermoFisher Scientific, Waltham, MA, USA) at equal density. The cell media was removed from the wells, then the wells were washed twice with DPBS, and 1 mL of phagocytosis assay medium containing 8 µL of pHrodo Red-conjugated synaptosomes was added into each well. The chambered coverglasses were kept in the incubator with 5% CO_2_ at 37 °C for 40 min, then the phagocytosis assay media was removed, and the wells were washed thrice with DPBS to remove any unbound pHrodo Red-conjugated synaptosomes. Then 800 µL of fresh phagocytosis assay medium containing lysotracker green DND-26 (1:20,000, Cell Signaling Technology, Danvers, MA, USA) was added into each well. After 6 h of incubation, the live imaging of cells was performed using a Nikon Eclipse Ti (Nikon, Tokyo, Japan).

### 2.12. RT-qPCR

Total RNA was isolated using the Aurum Total RNA Mini Kit (Bio-Rad, Hercules, CA, USA) according to the manufacturer’s protocol. The quantity and purity of the mRNA were analyzed using the NanoDrop ND-1000 Spectrophotometer (Thermo Fisher Scientific, Waltham, MA, USA). The complementary DNA (cDNA) was synthesized using the iScript cDNA synthesis Kit (Bio-Rad, Hercules, CA, USA). Amplification of cDNA was carried out with the CFX96 Touch Real-Time PCR Detection System (Bio-Rad, Hercules, CA, USA) using the SsoAdvanced Universal SYBR Green Supermix. The PCR protocol consisted of incubation for 2 min at 95 °C, followed by 40 cycles of amplification for 5 s at 95 °C and 30 s at 60 °C. Optimum primer pairs for genes of interest and glyceraldehydes 3-phosphate dehydrogenase (Gapdh), a housekeeping gene, were designed using Primer Express version 2.0.0 (Table 1).

### 2.13. Protein Extraction and Western Blotting

Glial cells (PMC and PAC) with 70–80% confluency were washed twice with ice-cold PBS and then lysed with MT Mammalian cell lysis buffer (Sigma-Aldrich, St. Louis, MO, USA) supplemented with a protease inhibitor cocktail. With the flask kept on ice, the cells were scraped and collected in micro-centrifuge tubes. Similarly, synaptosomes (syn) and different fractions (P1 (pellet 1), P2 (pellet 2), E (extrasynaptic fraction), S2 (cytosolic fraction)) collected during isolation of synaptosomes were also lysed with MT mammalian cell lysis buffer in the micro-centrifuge tubes, then the tubes were freeze/thawed in liquid nitrogen and centrifuged at 13,000 rpm for 30 min at 4 °C. The clear supernatant was collected, and the protein concentration was analyzed using BCA assay.

For Western blots, protein samples were prepared in a 1X SDS sample-loading buffer and denatured for 15 min at 85 °C. Equal amounts of protein (40 µg) were loaded onto a 10–12% tris–glycine polyacrylamide gel, run at 100 V in 1X running buffer and transferred from 10–12% tris–glycine polyacrylamide gel onto a PVDF membrane (activated in methanol) at 16 V for 60 min. The membranes were washed with TBST (10 mM Tris (pH 8.0), 150 mM NaCl and 0.01% Tween 20), blocked with 5% non-fat milk for 1 h, then washed thrice in TBST, and incubated overnight with primary antibodies to Iba1 (1:2000), GFAP (1:3000), PSD-95 (1:2000), Synaptophysin (1:3000) or β-actin (1:10,000). The membranes were washed five times with TBST, then incubated with mouse or rabbit HRP-conjugated secondary antibodies. A Chemiluminescent Imager (Thermo-Scientific) with a Supersignal West Femto Maximum Sensitivity Substrate was used to visualize the protein bands. Densitometry analysis was carried out utilizing Bio-Rad Quantity One image analysis software (Bio-Rad, Hercules, CA, USA).

### 2.14. Immunocytochemistry and Immunofluorescence of Brain Tissue

Cells cultured in coverslips/chamber slides were fixed for 30 min at room temperature in 4% paraformaldehyde, washed with PBS followed by methanol permeabilization for 30 min, blocked with serum containing 1% NGS and 3% BSA in PBS for 2 h at room temperature, and incubated overnight with primary antibodies at 4 °C with the following dilutions: Iba1 (1:1000), s100β (1:1000), GFAP (1:1000), NeuN (1:200), MBP (1:1000), Olig2 (1:1000), Cd11b (1:1000), and PSD95 (1:500). Following incubation, cells were washed five times with PBST (PBS + 0.1% Tween-20), and then incubated for 2 h at room temperature with a cocktail of secondary antibodies, all with 1:400 dilutions (Alexa Fluor 488 goat anti-chicken, Alexa Fluor 488 goat anti-rabbit, Alexa Fluor 488 goat anti-mouse, Alexa Fluor 546 goat anti-mouse, and Alexa Fluor 546 goat anti-rabbit). Coverslips/chamber slides were washed five times with PBST followed by mounting in VECTASHIELD medium with DAPI (Vector Laboratories, Burlingame, CA). Images were taken using a Leica confocal microscope SP8 (Leica Microsystems Inc., Buffalo Grove, IL, USA) or Nikon Eclipse TE2000-U inverted microscope (Nikon Instech Co. Ltd., Kawasaki, Kanagawa, Japan), and the Leica LAS X software (Leica Microsystems Inc.) or NIS-Elements microscope imaging software (Nikon Instech Co.).

For co-immunofluorescence of MAP2 and Iba1, the brains were collected, paraffin-embedded, and processed as described before [3]. The staining was performed using a combination of MAP2 (1:500) and Iba1 (1:500) overnight at 4 °C. Secondary antibodies were Alexa Fluor 488 goat anti-chicken and Alexa Fluor 546 goat anti-rabbit.

To quantify the association of microglia with neuronal cell bodies, images collected with 60x objective from normal and 22L-infected brains stained for Iba1 and MAP2 were merged and examined for instances of signal overlap. Neuronal cell bodies, in close proximity with microglia cell bodies or processes, were cropped and analyzed with colocalization plugin in ImageJ 1.52a. The area of signal overlap in normal (*n* = 21 measurements) and 22L-infected brains (*n* = 25 measurements) was plotted and compared with GraphPad PRISM, using an unpaired t-test with Welch’s correction.

### 2.15. Analysis of Gene Expression by Nanostring

After euthanasia by CO_2_ asphyxiation, the brains were immediately extracted and kept ice-cold during dissection. The brains were sliced using a rodent brain slicer matrix (Zivic Instruments, Pittsburg, PA). Cortex samples were collected from 2 mm central coronal sections of each brain. RNA isolation was performed as described previously [2]. RNA samples were processed by the Institute for Genome Science at the University of Maryland School of Medicine using the nCounter custom-designed Nanostring gene panel (Nanostring Technologies, Seattle, WA, USA), which consisted of genes that are expressed predominantly by astrocytes (www.brainrnaseq.org). Only samples with an RNA integrity number RIN > 7.2 were used for Nanostring analysis. All data passed quality control, with no imaging, binding, positive control, or CodeSet Content Normalization flags. The analysis of data was performed using nSolver Analysis Software 4.0. Ten house-keeping genes (Xpnpep1, Lars, Tbp, Mto1, Csnk2a2, CCdc127, Fam104a, Aars, Tada2b, and Cnot10) were used for the normalization of gene expression.

### 2.16. Statistics

Statistical analyses were performed using GraphPad PRISM 6 software (GraphPad software Inc., San Diego, CA, USA).

## 3. Results

### 3.1. Isolation of Astrocytes, Microglia, and Synaptosomes from Adult Mouse Brains

For examining phagocytic activity, astrocytes and microglia were isolated from adult animals according to the protocols that preserve glial reactive phenotypes [6]. Microglia and astrocytes were isolated from clinically sick C57Bl/6J mice (176–238 days post-inoculation, 220–283 days old) infected intraperitoneally with 22L mouse-adapted prion strain and age-matched C57Bl/6J control mice intraperitoneally injected with PBS.

The purity of primary microglia cultures was assessed using co-immunostaining for the microglial marker CD11b with the markers of oligodendrocytes (Olig2), astrocytes (GFAP), or neurons (NeuN). Ninety-five percent of cells were found to be CD11b-positive and Olig2-, GFAP-, and NeuN-negative (Figure 1A). Previously, we showed that primary microglia cultures originating from 22L-infected mice (22L-PMC) preserved a pro-inflammatory phenotype upon culturing in vitro, as illustrated by the elevated expression of proinflammatory genes (*Tnfα, Il1α, Il1β, Il10, Ccl2, Ccl4, Ccl6, Ccl9, Ccl12, Aif1, C1qa*, and *Tlr2*), when compared to primary microglia originating from age-matched controls (CT-PMC) [6]. Consistent with the previous work, Western blots showed the elevated expression of Iba1 in 22L-PMCs relative to CT-PMCs confirming that 22L-PMCs preserved their reactive phenotype (Figure 1B).

The purity of primary astrocyte cultures was assessed using co-immunostaining for the astrocyte marker GFAP with the markers of microglia (Iba1), neurons (NeuN), and oligodendrocytes (Olig2). Ninety-five percent of cells were found to be GFAP-positive and Olig2-, Iba1-, and NeuN-negative (Figure 2A). Moreover, co-immunostaining with GFAP and another astrocyte-specific marker, S100β, showed that the vast majority of cells were positive for both markers (Figure 2A). In our recent studies, primary astrocyte cultures originating from 22L-infected mice (22L-PACs) maintained the reactive phenotype as judged by the upregulation of the PAN-reactive markers Lcn2, Serpina3n, Steap4, Cxcl10, Hspb1, Vim, the A1 marker Serping1, the A2 marker S100a10, interleukins Il6, Il12b, Il33, and chemokine Ccl4, when compared to the primary astrocytes originating from age-matched controls (CT-PAC) [6]. Western blotting showed the elevated expression of GFAP in 22L-PACs relative to CT-PACs, confirming the reactive phenotype in 22L-PACs (Figure 2B).

For testing phagocytic activity, synaptosomes were isolated from clinically sick C57Bl/6J mice (176–238 days post-inoculation, 220–283 days old) infected with 22L prion strain and age-matched C57Bl/6J mice intraperitoneally injected with PBS. Western blot analysis confirmed that synaptosomes (syn) were enriched with postsynaptic and presynaptic markers PSD-95 and synaptophysin, respectively (Appendix A). Imaging using Transmission Electron Microscopy (TEM) showed morphological characteristics including synaptic vesicles (SV), synaptic junctions (SJ), post-synaptic density (PSD), and mitochondria (M) typical for preparations of synaptosomes (Appendix A).

### 3.2. Phagocytic Uptake of Synaptosomes Is Upregulated in Reactive Microglia

For examining phagocytic activity, purified synaptosomes were conjugated with pHrodo red followed by real-time, live-cell imaging of synaptosome uptake over a 24 h period using an Incucyte Live instrument. 22L-PMCs and CT-PMCs were each tested against synaptosomes isolated from 22L-infected animals (22L syn) and age-matched control animals (CT syn). In all samples, the phagocytic index, which represents a fraction of cells with fluorescent pHrodo red signal, reached a peak within the first 4 to 6 h followed by a decline (Figure 3A). Because pHrodo Red becomes fluorescent only in an acidic environment (pH < 6.3), synaptosomes that are simply attached to the cell but have not reached intracellular acidic compartments do not contribute to the phagocytic index. The decline in the phagocytic index after its peak suggests that the rate of intracellular degradation of synaptosomes exceeded the rate of their uptake.

Regardless of whether 22L syn or CT syn were used as a substrate of phagocytosis, 22L-PMCs displayed significantly higher phagocytic index in comparison to CT-PMCs (Figure 3A,B). Disease-associated 22L syn were phagocytosed at slightly higher rates relative to CT syn by both 22L-PMCs and CT-PMCs (Figure 3A–C). However, the differences in phagocytic indexes between 22L syn/22L-PMCs and CT syn/22L-PMCs, or 22L syn/CT-PMCs and CT syn/CT-PMCs reactions were not statistically significant (Figure 3A–C). These results illustrate that reactive microglia upregulate phagocytic activity toward synaptosomes regardless of whether they originate from normal or disease-affected animals. There is also a trend for more effective phagocytosis of 22L syn versus CT syn by both 22L-PMCs and CT-PMCs.

For examining whether phagocytosed synaptosomes are localized in intracellular acidic compartments, imaging of 22L-PMCs and CT-PMCs exposed to CT-syn using a Lysotracker, a fluorescent dye that tracks acidic compartments, was performed (Figure 4A,B). After 6 h of incubation, intracellular co-localization of green signal from the Lysotracker with the red fluorescence from pHrodo red-conjugated synaptosomes were observed in both 22L-PMCs and CT-PMCs confirming that synaptosomes are present in acidic phagolysosome (Figure 4A,B).

### 3.3. Phagocytic Uptake of Myelin Debris Is Upregulated in Reactive Microglia

For testing whether upregulation of phagocytic activity in reactive microglia is directed selectively toward synaptosomes as a substrate or generic, live-cell imaging using pHrodo Red-conjugated myelin debris was performed. Due to biosafety reasons, we could not use PrP^Sc^ as a substrate for phagocytosis. Similar to the results obtained with synaptosomes, the uptake of myelin debris was significantly elevated in 22L-PMCs relative to CT-PMCs (Figure 3D,E). Notably, upon reaching a peak at ~6 h, the phagocytic index declined at much slower rates in both 22L-PMCs/myelin debris and CT-PMCs/myelin debris reactions relative to the decline observed upon the uptake of synaptosomes by 22L-PMCs and CT-PMCs, respectively (Figure 3, compare A and D). Slower intracellular degradation of myelin debris in comparison to the degradation of synaptosomes offers one of the possible explanations for the different kinetics. Interestingly, in 22L-PMCs/myelin debris reaction, the phagocytic indexes were much lower than the indexes in 22L-PMCs/synaptosomes reactions, whether 22L-syn or CT-syn (Figure 3, compare A and D). The differences suggest that fewer cells were competent for the phagocytic uptake of myelin debris in comparison to the uptake of synaptosomes raising the possibility that different subpopulations of microglia are responsible for cleaning up different substrates. Nevertheless, this experiment illustrates that reactive microglia upregulate phagocytic activity regardless of the substrate.

### 3.4. Engulfment of Neuronal Cells by Reactive Microglia In Vivo

In vitro live-cell imaging suggested that synapses, and perhaps even neurons, are subject to phagocytotic engulfment and clearance by reactive microglia. For testing whether this is the case, the brains of animals infected with 22L were coimmunostained using antibodies toward a marker of microglia, Iba1, and neuronal protein MAP2. At the terminal stage of the diseases, reactive microglia were found in close proximity to neuronal cell bodies and often partially or fully engulfing them (Figure 5). Reactive microglia that engulf neuronal bodies were found across brain regions affected by 22L prion strain including hippocampus, thalamus, cortex, and brain stem (Figure 5A). In agreement with the recent studies [14], individual neuronal bodies were engulfed by single microglial cells. Very mild Iba1 staining was observed in age-matched control mice confirming its homeostatic, resting state (Figure 5B). Analysis of colocalization of Iba1 with MAP2 confirmed a much tighter association of microglia with neuronal cell bodies in 22L-infected animals relative to age-matched controls (Figure 5C).

### 3.5. Phagocytic Activity Is Reduced in Reactive Astrocytes

Having shown that phagocytic uptake is upregulated in reactive microglia, we asked whether the same was true for reactive astrocytes. The real-time phagocytic uptake of 22L and CT synaptosomes by 22L-PACs and CT-PACs was tested using live-cell imaging over a 24 h period. In all samples, the phagocytic index increased very slowly over 24 h suggesting that the rate of phagocytic uptake by astrocytes was much slower than that of microglia (Figure 6A). We do not know whether low levels of phagocytic uptake were, in part, due to fast rates of intracellular degradation, so that only a minuscule fraction of cells are fluorescent at any given time. Nevertheless, the slow rate of uptake by astrocytes was in agreement with previous studies that monitored the phagocytosis of neurons in live animals [14]. Interestingly, 22L-PACs displayed significantly lower phagocytic index relative to CT-PACs whether 22L or CT synaptosomes were used as a substrate (Figure 6A–C). Phagocytic activity of CT-PACs toward 22L syn was higher in comparison to activity toward CT syn, suggesting that normal astrocytes can discriminate between disease-associated and normal synaptosomes. 22L-PACs showed identical phagocytic indexes toward 22L and CT synaptosomes, which could be due to their extremely low phagocytic activity (Figure 6A). Nevertheless, the fraction of astrocytes competent for phagocytic uptake appeared to be considerably smaller than the fraction of phagocytose-competent microglia. Most importantly, in contrast to reactive microglia, phagocytic uptake was downregulated in reactive astrocytes.

To verify synaptosome phagocytosis using an independent approach, confocal microscopy imaging of PSD-95 and pHrodo Red was performed on fixed PACs upon their incubation with pHrodo Red-labeled synaptosomes. Colocalization of PSD-95 with pHrodo Red confirmed uptake and trafficking of synaptosomes into acidic intracellular compartments (Appendix A). In agreement with the above results, staining of PSD-95 showed a significant reduction in uptake of synaptosomes by 22L-PAC relative to that of CT-PAC (Appendix A). For illustrating that phagocytosed synaptosomes are localized with intracellular acidic compartments, 22L-PACs and CT-PACs incubated with pHrodo Red-labeled synaptosomes were imaged in the presence of LysoTracker. Again, intracellular colocalization of LysoTracker with pHrodo Red signal was observed in both 22L-PACs and CT-PACs confirming the presence of synaptosomes in acidic phagolysosomes (Figure 4C,D).

### 3.6. Downregulation of Phagocytic Activity Is Common in Reactive Astrocytes Associated with Prion Diseases

For testing whether the downregulation of phagocytic activity in reactive astrocytes is generic, live-cell imaging using pHrodo Red-conjugated myelin debris as a substrate was performed. Similar to the results obtained with synaptosomes, the uptake of myelin debris was significantly suppressed in 22L-PACs relative to CT-PACs (Figure 6D,E). Moreover, in a manner similar to synaptosome uptake, the phagocytic index grew very slowly over a 24-h period in both 22L-PACs and CT-PACs. These results suggested that the downregulation of phagocytic activity in reactive astrocytes is independent of a substrate, and supports the notion that the rate of phagocytic uptake in astrocytes is much slower than that in microglia.

Having shown that phagocytic uptake is downregulated in reactive astrocytes isolated from 22L-infected animals, next we asked whether the same was true for reactive astrocytes associated with other prion strains. To answer this question, the real-time phagocytic uptake of CT syn and CT myelin debris were performed using reactive astrocytes from mice infected with the mouse-adapted strain SSLOW [25,26]. This strain was selected because SSLOW-infected C57Bl/6J mice develop the disease with the shortest incubation time and exhibit very strong neuroinflammation across brain regions [3,21]. Again, the uptake of both CT syn and CT myelin debris was suppressed significantly in SSLOW-PACs relative to the uptake of corresponding substrates by CT-PACs isolated from age-matched control C57Bl/6J mice (Figure 7A–D). These results support the conclusion that phagocytic activity is suppressed in reactive astrocytes regardless of substrate or prion strain.

### 3.7. Analysis of the Expression of Genes That Report on Phagocytic Activity

Having shown that phagocytic uptake is up- and down-regulated in reactive microglia and astrocytes, respectively, next we asked whether the expression of genes associated with phagocytosis reflects the opposite trends. Indeed, when compared to CT-PMCs, 22L-PMCs showed upregulation in the expression of *CD68, Tlr2*, and *P2ry12* genes supporting earlier results that reactive microglia are primed for phagocytosis (Figure 8A). CD68 is a lysosomal receptor, which is involved in phagocytosis and often used as a phagocytic marker [29], P2ry12 is a chemoreceptor for adenosine diphosphate which senses a broad range of CNS insults and drives microglia into a phagocytic state [30,31,32], whereas Tlr2 is a Toll-like receptor that activates signaling responsible for phagocytosis [33].

In astrocytes, two pathways, which are associated with Mertk and Abca1, were shown to be involved in phagocytosis [16]. In agreement with the results on live-cell imaging, several genes associated with Mertk and Abca1 pathways (*Mertk, Mfge8, Elmo3, Gulp1, Megf10*) were downregulated in 22L-PACs when compared to CT-PACs (Figure 8B). Previously, MEGF10 (multiple EGF-like-domains 10) and Gulp1 (engulfment adapter phosphotyrosine-binding domain containing 1) were found to be important for the phagocytosis of synapses and apoptotic cells by astrocytes [16,34]. Mertk functions as an engulfment receptor and is also involved in the clearance of apoptotic cells, whereas Mfge8 works as a bridge and attaches apoptotic cells to phagocytes [35].

Analysis of gene expression in bulk brain tissues of mice infected with 22L revealed that all three genes (*CD68, Tlr2, P2ry12*) that were associated with phagocytic activity in microglia and upregulated in 22L-PMCs, were also upregulated in animals (Figure 8C). Since all three genes are microglia-specific, this result supports the conclusion that phagocytic uptake is activated in reactive microglia associated with prion diseases. Among the genes associated with the phagocytosis of astrocytes, Mertk was upregulated, whereas Mfge8 was downregulated in animals infected with 22L. The contrasting results for Mertk expression in animals and 22L-PACs can be attributed to the lack of cell-specificity for this gene, which is expressed by microglia too.

## 4. Discussion

Little is known about the phagocytic activity of reactive microglia and astrocytes associated with prion diseases. Astrocyte and microglia cell lines, as well as primary astrocyte cultures, can phagocytose PrP^Sc^ in vitro [18,19,20]. Moreover, PrP^Sc^ aggregates were found in reactive astrocytes and microglia in animals infected with prions [21,22,23,24]. However, whether phagocytic activities are up- or downregulated in the reactive states of astrocytes and microglia has never been tested.

The current study is the first to examine the phagocytic activity of reactive astrocytes and microglia isolated from prion-infected animals. We found that the reactive microglia upregulate, whereas the reactive astrocytes downregulate phagocytic uptakes. The up- and downregulation of phagocytosis by the two cell types were observed irrespective of the substrates tested (disease-associated or normal synaptosomes or myelin debris) indicating that the dysregulations are dictated by the reactive states, not substrates. Consistent with previous studies [14], the kinetics of phagocytic uptake by microglia was faster relative to that of astrocytes. Moreover, in agreement with the traditional view that microglia are professional phagocytes, a significantly higher fraction of microglia cells were competent in phagocytic uptake in comparison to astrocytes. In recent studies that performed the imaging of phagocytosis in live animals under a normal homeostatic environment, microglia and astrocytes were found to share responsibilities in their phagocytic uptake coordinating the space, time, and substrate [14]. Microglia were found to specialize in engulfing cell bodies, dendrites, and nuclei, whereas astrocytes were responsible for the clearing of small diffuse apoptotic fragments derived from distal neuronal branches or dendritic arbors [14]. In this study, reverse changes of the phagocytic activities in reactive microglia and astrocytes reveal an imbalance in this important homeostatic function.

This study demonstrated the upregulation of phagocytic activity in reactive microglia with respect to both disease-associated and normal synaptosomes. In fact, the rate of phagocytic uptake of normal synaptosomes was almost as high as the uptake of disease-associated synaptosomes suggesting that in its reactive states, microglia are primed for the phagocytosis of functional synapses and, perhaps, even viable neurons that do not present disease-associated cues. In support of the notion that reactive microglia can be neurotoxic, the staining of 22L-infected brain slices revealed reactive microglia to be in close proximity to, or engulfing, neuronal cell bodies across brain regions affected by prions (Figure 5). Moreover, 22L-infected animals showed the upregulation of genes associated with reactive microglia along with genes involved in phagocytosis (*P2ry12, CD68, Tlr2*). P2RY12 is a receptor that senses CNS insults and drives microglia into a phagocytic state [30,31,32], Tlr2 is a receptor that recognizes multiple pathogens, including prions [36,37], and CD68 is a lysosomal receptor closely associated with phagocytosis [29]. Together, these results highlight the neurotoxic potential of reactive microglia associated with prion disease. The neurotoxic mechanisms involving synapse elimination by reactive microglia received solid support in studies on Alzheimer’s disease, frontotemporal dementia, as well as normal aging [38,39,40,41]. Key components of the synapse pruning machinery include C1q, which is involved in tagging synapses, and C3, which drives engulfment and phagocytosis via interaction with its receptor C3ar1, expressed by microglia [42,43]. In agreement with this mechanism, the key components of a complement cascade including *C1qa, C1qb, C1qc*, *C3*, and *C3ar1* were found to be upregulated in prion-infected mice [2,21,44].

The current studies support the neurotoxic mechanism on the elimination of synapses and neurons by microglia, at least at the late stages of the disease, which appears to contradict mounting evidence that in prion diseases, microglia are neuroprotective [10,11]. Indeed, partial ablation of microglia by PLX5622 accelerated disease progression, highlighting the net positive impact of microglia [10]. Moreover, microglia depletion led to the increased deposition of PrP^Sc^ in organotypic cultured slices and shortened the incubation time for the disease in mice [11]. It is assumed that the main neuroprotective mechanism involves phagocytic clearance of PrP^Sc^. Due to biosafety reasons, we were not able to examine the phagocytic clearance of PrP^Sc^ in the current studies. Nevertheless, consistent with the mechanism on phagocytic clearance, PrP^Sc^ aggregates were found in the reactive microglia of animals infected with prions in previous studies [21,22,23,24]. Moreover, the activation of microglia by viral infection was found to stimulate the transient clearance of PrP^Sc^ in mice infected with the RML prion strain, although, transient clearance did not change the incubation time for the disease [45]. Alternatively, in mice infected with ME7, reactive microglia were found to upregulate their capacity to clear beads and apoptotic cells but were not effective in cleaning PrP^Sc^ [17], contradicting the neuroprotective hypothesis. The activation of microglia along with the accumulation of PrP^Sc^ was found to be necessary for neuronal cell death [46]. In support of the neurotoxic hypothesis, knockout of Cx3cr1, a receptor of the Cx3cl1 chemokine that is important for maintaining the homeostatic resting state of microglia, reduced incubation times in mice challenged with RML, ME7, or MRC2 strains [47].

For reconciling the neuroprotective and neurotoxic hypothesis, the following possibilities should be considered: (i) the role of microglia might change with the progression of the diseases, and (ii) the microglia phenotype and degree of activation are dictated, in part, by strain-specific features. The first possibility, i.e., an evolution of the microglia reactive phenotypes with the disease progression, assumes that the primarily protective phenotype at early disease stages, which is characterized by effective phagocytosis of PrP^Sc^, gives rise to the overactivated, predominantly neurotoxic phenotype at the later stages, which targets synapses and viable neurons. As an example, in animal models of AD, microglia were found to be active in phagocytic uptake of A*_β_* peptides at the early stages of the disease [48,49]. However, overwhelming phagocytic uptake along with age-related changes can burden the phagocytic functions of microglia causing senescence at older ages along with impairments in phagocytic activity [50,51,52]. The second possibility involves the effects of strain-specific features, such as sialylation levels of PrP^Sc^ glycans [53,54], on the activation of phagocytic pathways and dictating microglia reactive states. Sialylation of glycans acts as a part of a self-associated molecular pattern helping the innate immunity to recognize “self” from “altered self” or “non-self” [55,56]. The stripping of sialic acid residues exposes galactose residues that generate “eat me” signals for professional and non-professional macrophages including microglia. Prion strains are different with respect to their sialylation status [53,57,58]. In support of the hypothesis on the role of sialylation in defining the phagocytic state of microglia, the partial stripping of sialic acid residues helped to clear prion infection in an organism [59,60,61]. Moreover, studies that employed primary cultures showed that de-sialylation of PrP^Sc^ enhanced the proinflammatory response in microglia [62]. In animals, prion strain with a low level of sialylation was found to induce a very strong proinflammatory response, where PrP^Sc^ was found to primarily colocalize within microglia [21].

In contrast to microglia, the phagocytic activities of reactive astrocytes were found to be downregulated in this study. Phagocytic uptake was suppressed regardless of the phagocytic substrate and observed in 22L- and SSLOW-associated astrocytes, suggesting that downregulation is a common feature of the reactive phenotype associated with prion diseases. The results on reduced phagocytic uptake seen in real-time imaging were supported by the downregulation of the expression of genes associated with phagocytosis (*Mertk, Mfge8, Elmo3*, *Gulp1*, *Megf10*) (Figure 8B). Since the expression of *Mfge8* in vivo is primarily limited to astrocytes [63], the downregulation of *Mfge8* in bulk brain tissues of 22L-infected animals in the current work suggested that the phagocytotic activity of astrocytes is also suppressed in vivo (Figure 8C). In previous studies, the ablation of *Mfge8* reduced the clearance of apoptotic bodies, enhanced the accumulation of PrP^Sc^, and accelerated disease progression [63], suggesting that *Mfge8*-dependent pathways play an important neuroprotective role. Consistent with previous findings [3,6,63], the current study illustrates that *Mfge8* is downregulated in animals, which leads to the suppression of phagocytic activity in reactive astrocytes and contributes to their neurotoxic phenotype.

Changes in the phagocytotic activity of glia can be beneficial or detrimental depending on the substrate of phagocytosis [8]. Phagocytic clearance of PrP^Sc^, along with apoptotic bodies and cell debris, is neuroprotective, whereas the elimination of synapses, along with viable neurons, contributes to neurotoxicity. For the reasons outlined below, predicting the net impact of up- or downregulation of phagocytic activity in microglia and astrocytes might be difficult. First, it is not clear whether glial phagocytic activity can be regulated selectively to target only a subset of phagocytic substrates, i.e., those involved exclusively in neurotoxic or neuroprotective mechanisms. Limited evidence has been put forward that distinct subpopulations of microglia that are active toward different substrates exist [64]. Second, taking into consideration the asynchronous progression of the disease in different brain regions along with selective tropism of prion strains to different regions [2,3,22,65,66], it would not be surprising if distinct phagocytic states of microglia and astrocytes exist at any given time point in different brain regions. Third, as mentioned above, phagocytic states of glia might be dictated by strain-specific features.

In this current work, phagocytic uptakes by astrocytes and microglia were tested separately. However, in vivo, their phagocytic activities are tightly coordinated via cross-talk between the two cell types that rely on multiple pathways [14]. How does cross-talk between microglia and astrocytes affect their phagocytic functions? In the absence of tyrosine kinase receptor Mertk, which is expressed by both microglia and astrocytes, the engulfment of cell bodies by microglia was delayed, whereas astrocytes failed to polarize toward dying cells [14]. Reactive astrocytes including those associated with prion diseases upregulate expression of IL-33 and C3 [6], which are known to drive microglia-mediated synapse engulfment and elimination [42,43,67]. Mgfe8 opsonizes apoptotic bodies along with other targets and mediates their phagocytic clearance by binding to the Mgfe8 receptor on microglia, astrocytes, and other phagocytes [35,68]. We cannot exclude that very low rates of phagocytic uptake in astrocyte culture were in part attributed to the lack of microglial factors important for the phagocytic activity of astrocytes.

To summarize, this study illustrates that in the reactive states, the phagocytic activities of astrocytes and microglia are dysregulated in opposite directions. Considering that the phagocytosis of damaged neurons involves coordinated efforts by both microglia and astrocytes [14], an imbalance in phagocytic uptake between reactive microglia and astrocytes can lead to excessive microglia-mediated neuronal death, yet disabled removal of cell debris by astrocytes, contributing to the neurotoxic effects of glia as a whole. This work raises several questions which would be interesting to address in future studies. It is not known whether changes in phagocytic activities are dictated by specific brain areas, i.e., controlled by site-specific microenvironments. It is also of great interest whether phagocytosis could be selectively up- or downregulated for specific substrates. The mechanisms responsible for cross-talk between astrocytes and microglia that coordinates their phagocytic activity are also important to investigate.

## Figures and Tables

**Figure 1 cells-10-01728-f001:**
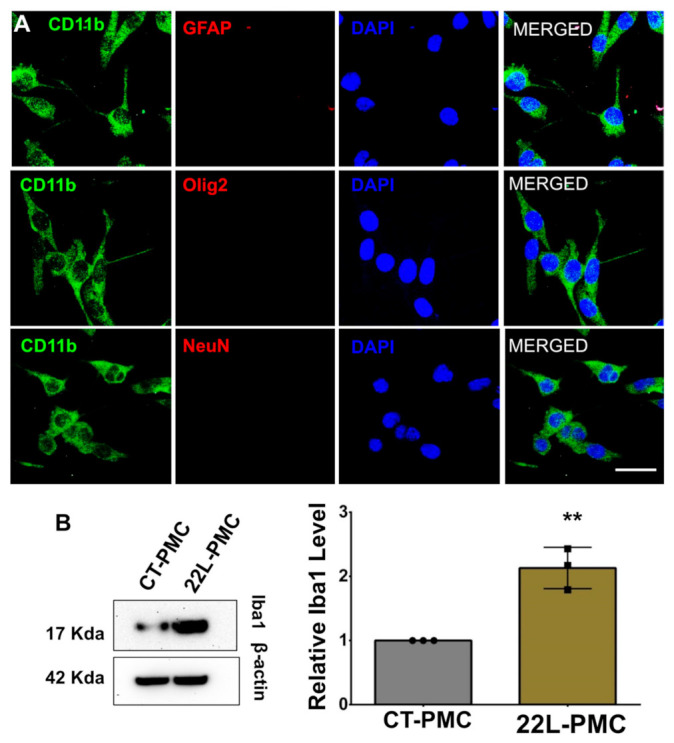
Preparation of adult primary microglia cell cultures. (**A**) Co-immunostaining of PMCs for microglia (CD11b, green) and astrocytes (GFAP, red), oligodendrocytes (Olig2, red), or neurons (NeuN, red). Cell nuclei are stained with DAPI (blue). Images are representatives of three independent cultures, each from an individual control animal. Scale bars = 25 μm. (**B**) Representative Western blots (left) and densitometric analysis of Iba1 expression levels (right) in CT-PMCs and 22L-PMCs normalized per expression of β-actin. Data are expressed as mean ± SEM, *n* = 3 independent cultures isolated from individual animals, ** *p* < 0.01 (two-tailed, unpaired student t-test).

**Figure 2 cells-10-01728-f002:**
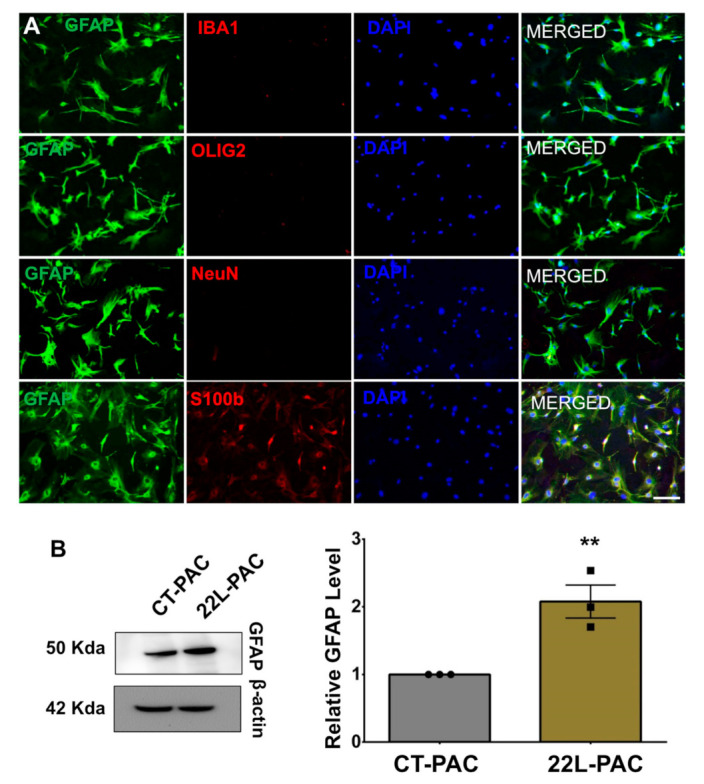
Preparation of adult primary astrocyte cell cultures. (**A**) Co-immunostaining of PACs using an astrocyte-specific marker (GFAP, green) and microglia- (Iba1, red), oligodendrocyte- (olig2, red), neuron- (NeuN, red) or second astrocyte- (S100β, red) specific marker. Cell nuclei are stained with DAPI (blue). Images are representatives of three independent primary cell cultures, each prepared from an individual control animal. Scale bar = 50 μm. (**B**) Representative Western blots (left) and densitometric analysis of GFAP expression levels (right) in CT-PACs and 22L-PACs normalized per expression of β-actin. Data are expressed as mean ± SEM, *n* = 3 independent cultures isolated from individual animals, ** *p* < 0.01 (two-tailed, unpaired student t-test).

**Figure 3 cells-10-01728-f003:**
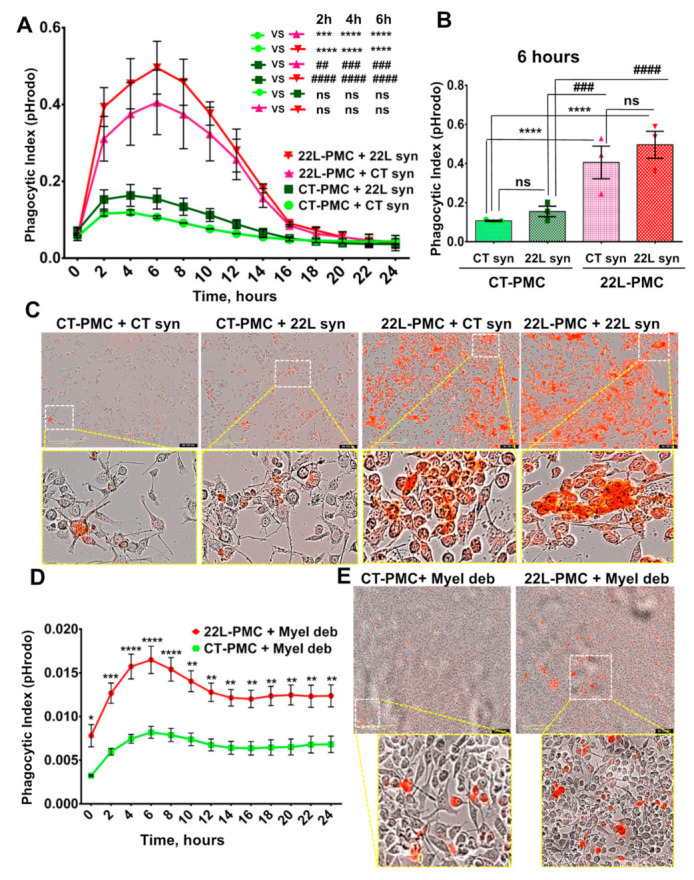
Live-cell imaging of phagocytic activity of PMCs. (**A**) Real-time, live-cell imaging of uptake of 22L-syn or CT-syn by 22L-PMCs or CT-PMCs monitored by pHrodo Red fluorescence assay over the time period of 24 h. (**B**) Analysis of phagocytic indexes measured by live-cell pHrodo Red fluorescence assay at the 6-h time point. (**C**) Representative overlaps of phase-contrast and fluorescence images of 22L-PMCs and CT-PMCs exposed to 22L-syn or CT-syn collected at the 12 h time point. (**D**) Real-time, live-cell imaging of uptake of myelin debris by 22L-PMCs and CT-PMCs over the time period of 24 h monitored by pHrodo Red fluorescence assay. (**E**) Representative overlaps of phase-contrast and fluorescence images of 22L-PMCs and CT-PMCs exposed to myelin debris. For all live-cell imaging assays, *n* = 3 22L-PMC or CT-PMC cultures, each isolated from an individual animal and cultured in three wells. For each well, four images were automatically acquired every 2 h. *n* = 3 independent isolations from individual animals for each 22L-syn, CT-syn, or myelin debris. Data presented as mean ± SEM. * *p* ≤ 0.05, ** or ^##^ *p* ≤ 0.01, *** or ^###^ *p* ≤ 0.001 and **** or ^####^ *p* ≤ 0.0001, and ‘ns’ non-significant by a two-way ANOVA followed by Tukey’s test.

**Figure 4 cells-10-01728-f004:**
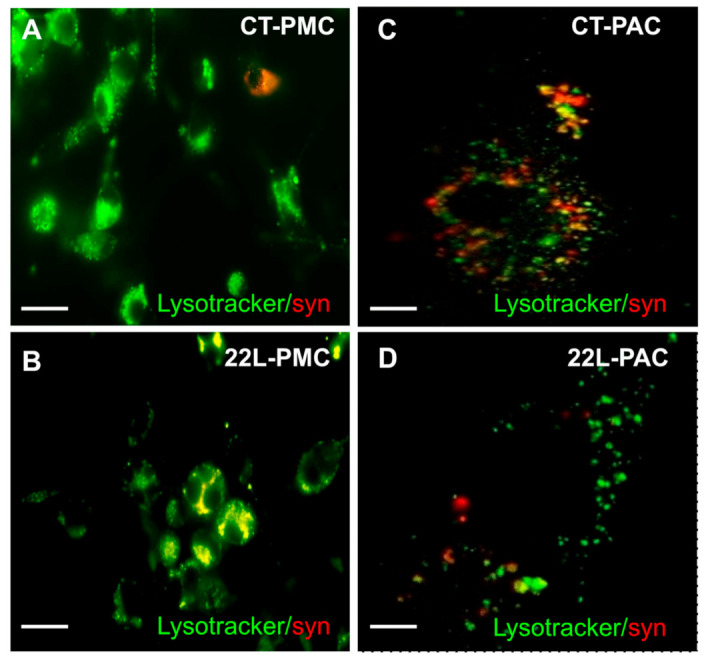
Imaging of phagocytic activity using LysoTracker assay. Representative images of CT-PMCs (**A**), 22L-PMCs (**B**), CT-PACs (**C**), and 22L-PACs (**D**) exposed to pHrodo Red-conjugated CT-syn (red) and incubated for 6 h in the phagocytic assay media containing LysoTracker Green. Scale bars = 50 µm (**A**,**B**) and 10 µm (**C**,**D**).

**Figure 5 cells-10-01728-f005:**
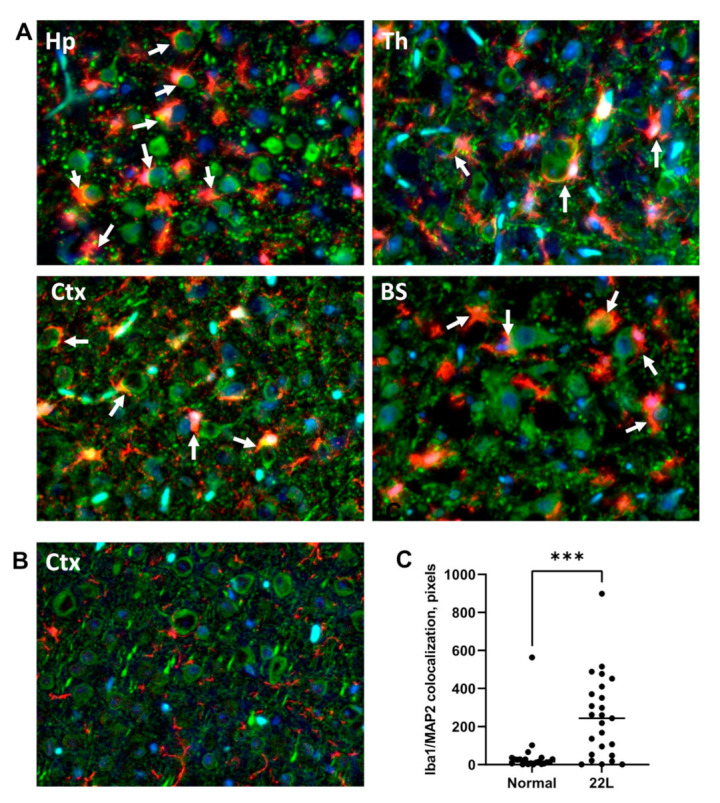
Engulfment of neuronal cells by reactive microglia in vivo. Co-immunostaining of 22L-infected mice (**A**) and age-matched controls (**B**) for microglia (Iba1, red), neurons (MAP2, green), and nuclei (DAPI, blue). In 22L-infected animals, microglia are often found in close vicinity to neuronal cell bodies or engulfing neuronal cell bodies, as pointed by white arrows. (**C**) Colocalization analysis of Iba1 and MAP2 in cortex. *** *p* < 0.001 (by unpaired t*-*test with Welch’s correction). Hp–hippocampus, Th–thalamus, Ctx–cortex, BS–brain stem. Scale bar: 50 μm.

**Figure 6 cells-10-01728-f006:**
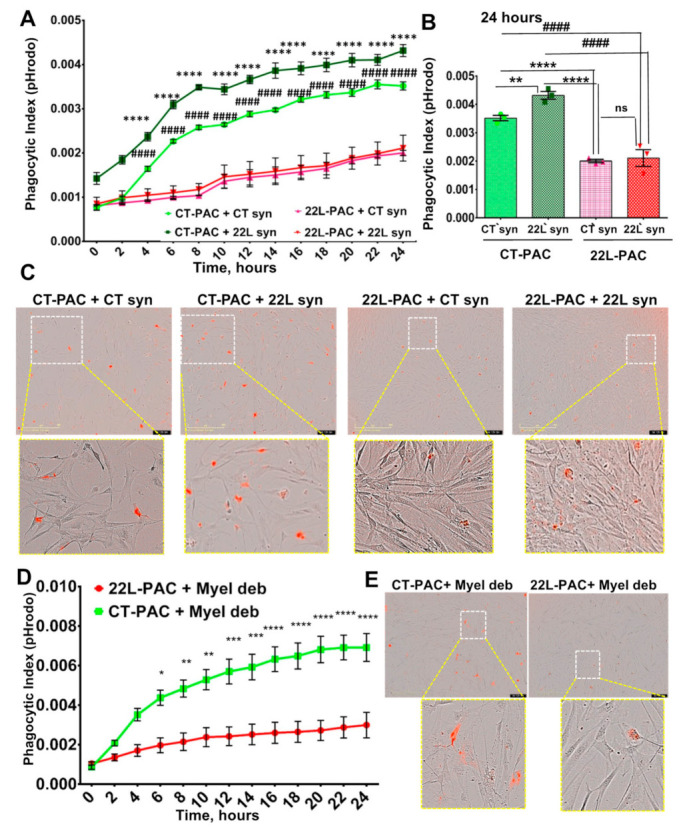
Live-cell imaging of phagocytic activity of PACs from 22L-infected mice. (**A**) Real-time, live-cell imaging of uptake of 22L-syn or CT-syn by 22L-PACs or CT-PACs monitored by pHrodo Red fluorescence assay over the time period of 24 h. (**B**) Analysis of phagocytic indexes measured by live-cell pHrodo Red fluorescence assay at 24 h time point. (**C**) Representative overlaps of phase-contrast and fluorescence images of 22L-PACs and CT-PACs exposed to 22L-syn or CT-syn. (**D**) Real-time, live-cell imaging of uptake of myelin debris by 22L-PACs and CT-PACs over the time period of 24 h monitored by pHrodo Red fluorescence assay. (**E**) Representative overlaps of phase-contrast and fluorescence images of 22L-PACs and CT-PACs exposed to myelin debris. For all live-cell imaging assays, *n* = 3 22L-PAC or CT-PAC cultures, each isolated from an individual animal and cultured in three wells. For each well, four images were automatically acquired every 2 h. *n* = 3 independent isolations from individual animals for each 22L-syn, CT-syn, or myelin debris. Data presented as mean ± SEM. * *p* ≤ 0.05, ** *p* ≤ 0.01, *** *p* ≤ 0.00,1 and **** or ^####^ *p* ≤ 0.0001, and ‘ns’ non-significant by a two-way ANOVA followed by Tukey’s test.

**Figure 7 cells-10-01728-f007:**
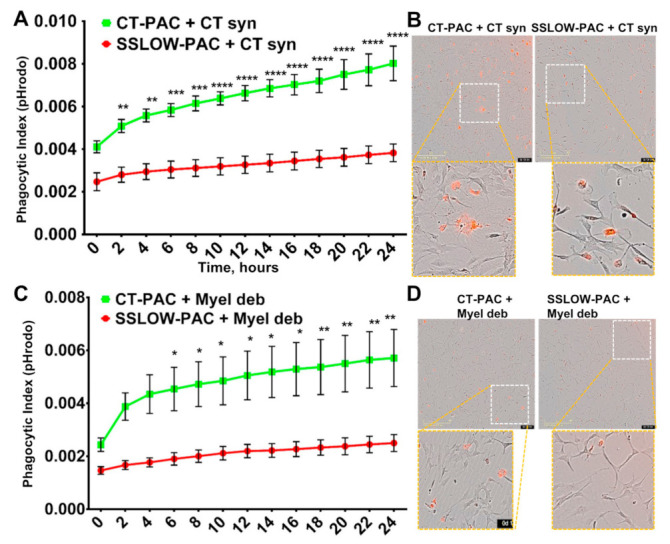
Live-cell imaging of phagocytic activity of PACs from SSLOW-infected mice. (**A**) Real-time, live-cell imaging of uptake of CT-syn by SSLOW-PACs or CT-PACs monitored by pHrodo Red fluorescence assay over the time period of 24 h. (**B**) Representative overlaps of phase-contrast and fluorescence images of SSLOW-PACs and CT-PACs exposed to 22L-syn or CT-syn. (**C**) Real-time, live-cell imaging of uptake of myelin debris by SSLOW-PACs and CT-PACs over the time period of 24 h monitored by pHrodo Red fluorescence assay. (**D**) Representative overlaps of phase-contrast and fluorescence images of SSLOW-PACs and CT-PACs exposed to myelin debris. For all live-cell imaging assays, *n* = 3 SSLOW-PAC or CT-PAC cultures, each isolated from an individual animal and cultured in three wells. For each well, four images were automatically acquired every 2 h. *n* = 3 independent isolations from individual animals for each CT-syn or myelin debris. Data presented as mean ± SEM. * *p* ≤ 0.05, ** *p* ≤ 0.01, *** *p* ≤ 0.001, and **** *p* ≤ 0.0001 by a two-way ANOVA followed by Tukey’s test.

**Figure 8 cells-10-01728-f008:**
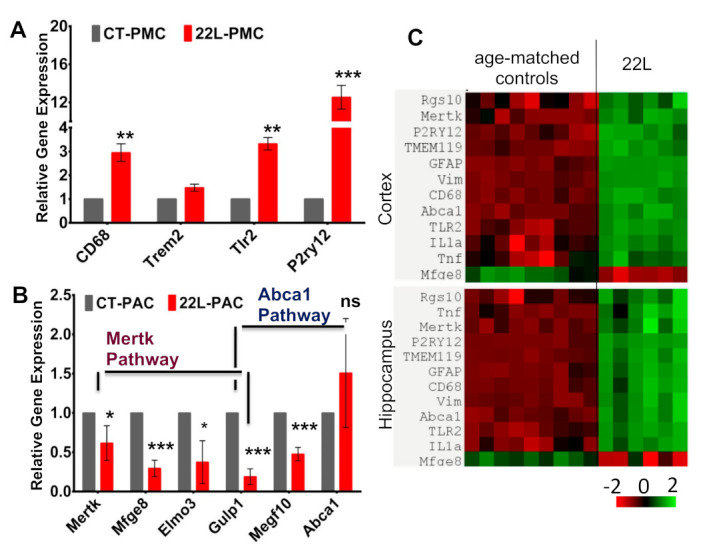
Analysis of gene expression. (**A**,**B**) Expression of genes associated with phagocytosis in 22L-PMCs normalized by the expression levels in CT-PMCs (**A**), or 22L-PACs normalized by the expression levels in CT-PACs (**B**) analyzed by qRT-PCR. Gapdh was used as the housekeeping gene. Data represent mean ± SEM, *n* = 3 independent cultures isolated from individual animals, each analyzed in triplicates, * *p* ≤ 0.05, ** *p* ≤ 0.01, *** *p* ≤ 0.001, and ‘ns’ non-significant by a two-tailed, unpaired student t-test. (**C**) Heat map of gene expression in cortex and hippocampus of 22L-infected and age-matched control mice (*n* = 6 animals per group).

**Table 1 cells-10-01728-t001:** Primer sequences for qRT-PCR.

Primer	Accession Number	Sequence
CD68	NM_001291058.1	F 5′- CTGCCAGTCCGAAAATGGAAC-3′ R 5′- CTTCATCCACCGGGGCTATC-3′
P2ry12	NM_027571.4	F 5′- AACACCACCTCAGCCAATAC-3′ R 5′- ACAGCAATGGGAAGAGAACC-3′
Trem2	NM_031254.3	F 5′- TGGGACCTCTCCACCAGTT-3′ R 5′- GTGGTGTTGAGGGCTTGG-3′
Tlr2	NM_011905.3	F 5′- CACTATCCGGAGGTTGCATATC-3′ R 5′- GGAAGACCTTGCTGTTCTCTAC-3′
Abca1	NM_013454.3	F 5′- ATGGAGCAGGGAAGACCAC-3′ R 5′- GTAGGCCGTGCCAGAAGTT-3′
Megf10	NM_001001979.2	F 5′- CTACAGACACAAGCAGAAGAGG-3′ R 5′- CAGGGTTTCTGCGATGGTATAG-3′
Mfge8	NM_008594.2	F 5′- GTGCCCTGTGGGCTACTC-3′ R 5′- GTATTGGGGACGGCTGTG-3′
Mertk	NM_008587.2	F 5′- GATGGTTCTGGCCCCACT-3′ R 5′- CTGATCTAGCTCGGTCTCTTCC-3′
Elmo3	NM_17‘2760.3	F 5′- CTCAGCACTGCCCCAGAT-3′ R 5′- GCACCCTTCACGACTTGTATT-3′
Gulp1	NM_028450.3	F 5′- GGATTAGAAGGAGGGAGAGGA-3′ R 5′- ACGATCGCGTGTTCGTATC-3′
Gapdh	NM_001289726.1	F 5′- AACAGCAACTCCCACTCTTC-3′ R 5′- CCTGTTGCTGTAGCCGTATT-3′

## Data Availability

All the data presented in this study are included in this article.

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
