# Peer review of "Phagocytic Activities of Reactive Microglia and Astrocytes Associated with Prion Diseases Are Dysregulated in Opposite Directions"

_cells, 2021, doi:10.3390/cells10071728_

Round 1
Reviewer 1 Report
In the present study, Sinha et al., demonstrate the phagocytic activity of both reactive microglia and reactive astrocytes in a model of Prion disease. Using primary cell cultures along with qPCR and additional approaches, the authors showed a significant increase in microglia phagocytic activity and downregulation of phagocytic activity in astrocytes. The authors suggest that these differences drive increases in neurotoxicity as it drives microglia cells to phagocytize neuronal cell bodies.
The data presented in the study is sound and the experimental approached appropriate for the goals of the study. The only weakness of the study is the lack of dynamic in vivo correlates to show that the up and down regulation of phagocytic activity between microglia and astrocytes is as observed in cell culture and suggested by the molecular studies. Alternatively, inclusion of immunohistochemistry from 22L-infected brain would be important. Overall, an interest study with important findings.
Author Response
We thank the reviewers for his/her positive comments and appreciation of our work. We fully agree that examining phagocytic activity of astrocytes and microglia in live animals would be important step forward. This task requires tools and equipment devoted 100% for working with prion-infected animals, which is currently outside of our reach. Unfortunately, shared equipment or university facilities cannot be used for examining prion-infected animals for biosafety reasons. Immunochemistry of 22L-infected brain is presented in figure 5.
Reviewer 2 Report
The current study by Sinha et al presents an examination of the phagocytic ability of microglia and astrocytes isolated from a prion infected mouse line. Through phagocytic assays, the authors demonstrated that microglia have an elevated phagocytic response while astrocytes have a decreased phagocytic response. Overall, this is an interesting study that takes a look at an important part of disease biology. Although the results are interesting and the discussion is well presented, there are a couple areas which could still be improved.
- Figure 6 should include quantitation to back up claims that there is more engulfment of neurons by microglia in vivo found in the prion infected mouse. The number of Iba1+/Map+ colocalization can be compared between the 22L-infected mice and control mice. It would be a good validation of the in vitro results.
- Figure 3 can probably be supplementary. It does not really offer much for the results section.
- In almost all of the figures, the appearance is quite sloppy. Figure image panels are not lined up, lines of text are not properly aligned, lines that should be straight are slightly angled, and there is different sizes of white spaces between each panel. The authors need to clean up each figure and align everything better.
Author Response
We thank the reviewers for his/her positive comments, appreciation of our work, and constructive critical points.
Figure 6 (current Figure 5). It is difficult to rely on Iba1+/Map+ colocalization for objective quantification of the engulfment of neurons by microglia, because in most cases, reactive microglia encircle or surround neuronal bodies while showing limited if any colocalization between Iba1 and Map. Simply counting a microglia cells is also not an option, because of difficulties in identifying individual microglia cells in age-matched controls due to limited expression of Iba1 in homeostatic microglia (Figure 5B).
As recommended, Figure 3 is now moved to supplementary material.
Per reviewer advise, the presentation of figures has been cleaned up and improved.
Round 2
Reviewer 2 Report
In this revised manuscript, the authors have elected to not perform the suggested in vivo counts of phagocytosing microglia. Quantitative analysis is important if the authors want to demonstrate a significant phagocytic difference occurs in vivo. Qualitative images are not sufficient. Furthermore, the assertion that Iba1 is a reactive marker for microglia is incorrect. Iba1 should stain all microglia and macropahges present in the tissue. Iba1 should be visible even in homeostatic microglia. If the authors still elect not to perform this analysis, the language should be changed to reflect this, in addition to adding more text to discussion clarifying this limitation. Furthermore, Iba1 should not be listed as a reactive microglia marker.
Several of the figures still have not been fixed to remove the sloppy appearance. In figure 1&2, the image boxes need to be lined up evenly and the amount of white space between panels should be the same. The same could be said for almost every image box. It should be simple in whatever program being used to create figures to automatically line up image boxes and text lines to give the figures a professional look.
Author Response
We thank the reviewers for his/her constructive comments aiming at improving the manuscript.
As requested by the reviewer, analysis of colocalization of microglia and neurons are now shown in new panel C in Figure 5, which confirms much tighter association of microglia with neuronal cell bodies in 22L-infected animals relative to the age-matched controls (Fig. 5C). As we indicated in the previous response, the colocalization analysis underestimate the actual level of engulfment in 22L-animals, because in many cases, reactive microglia encircle or surround neuronal bodies while showing limited colocalization between Iba1 and Map. Nevertheless, the analysis confirms tight association of microglia and neuronal bodies in 22L-infected animals.
We apologize, if our response in previous comment regarding limited expression of Iba1 in homeostatic microglia was misunderstood. We are well aware that Iba1 is expressed by reactive and homeostatic states. We meant that identification of individual microglia cells in control animals would be difficult due to highly ramified morphology of homeostatic microglia. Identification of individual cells would require 3D reconstruction of confocal microscopy images, which is not available to us.
Figure 1 and 2 are replaced and all panels are now lined up.